# Intrinsic multiplication rate variation and plasticity of human blood stage malaria parasites

Lindsay B. Stewart[1], Ofelia Diaz-Ingelmo[1], Antoine Claessens [1,2], James Abugri[3,4], Richard D. Pearson [5,6], Sonia Goncalves[6], Eleanor Drury [6], Dominic P. Kwiatkowski [5,6], Gordon A. Awandare[4] & David J. Conway [1✉]

Pathogen multiplication rate is theoretically an important determinant of virulence, although often poorly understood and difficult to measure accurately. We show intrinsic asexual blood stage multiplication rate variation of the major human malaria parasite *Plasmodium falciparum* to be associated with blood-stage infection intensity in patients. A panel of clinical isolates from a highly endemic West African population was analysed repeatedly during five months of continuous laboratory culture, showing a range of exponential multiplication rates at all timepoints tested, mean rates increasing over time. All isolates had different genome sequences, many containing within-isolate diversity that decreased over time in culture, but increases in multiplication rates were not primarily attributable to genomic selection. New mutants, including premature stop codons emerging in a few isolates, did not attain suffi-ciently high frequencies to substantially affect overall multiplication rates. Significantly, multiplication rate variation among the isolates at each of the assayed culture timepoints robustly correlated with parasite levels seen in patients at clinical presentation, indicating innate parasite control of multiplication rate that contributes to virulence.

[1] Department of Infection Biology, London School of Hygiene and Tropical Medicine, Keppel St, London WC1E 7HT, UK. [2] LPHI, MIVEGEC, INSERM, CNRS, IRD, University of Montpellier, 34095 Montpellier, Cedex 5, France. [3] Department of Applied Chemistry and Biochemistry, Faculty of Applied Sciences, University for Development Studies, Tamale, Ghana. [4] West African Centre for Cell Biology of Infectious Pathogens (WACCBIP) and Department of Biochemistry, Cell and Molecular Biology, University of Ghana, Legon, Accra, Ghana. [5] MRC Centre for Genomics and Global Health, Big Data Institute, University of Oxford, Oxford OX3 7BN, UK. [6] Wellcome Sanger Institute, Cambridge CB10 1SA, UK. ✉email: david.conway@lshtm.ac.uk

  **1**

t is vital to understand blood-stage multiplication rates of *Plasmodium falciparum*, as this single malaria parasite species causes more human mortality than all other eukaryotic pathogen species combined[1]. It adapted specifically to humans over the past several thousand years, after having been derived as a spill-over zoonosis from an ancestral parasite species infecting gorillas[2]. Clinical severity of *P. falciparum* malaria is correlated with overall numbers of parasites in the blood[3,4], and appears to be related to multiplication rates of parasites in patients as estimated by modelling[5].

The first estimates of *P. falciparum* multiplication rates were derived from induced infections of volunteers or neurosyphilis patients subjected to pyretic 'malariatherapy' in different institutions between the 1920s and the 1950s[6,7]. A few different parasite strains were used, variation among them being indicated when combined data were later analysed, although methodological differences among institutions may confound these retrospective comparisons[6,7]. Recent studies of experimental infections in volunteers allow more accurate estimates of initial *in vivo* multiplication rates, although these vary slightly depending on the inoculation method[8–12] as well as the statistical methods applied[11,13], and variation among parasites has not been focused on given that a single strain has been mainly used (comparisons with an unrelated strain indicated differences in one study[14] but not another[15]).

Assays of parasites from patients during the first *ex vivo* cycle in laboratory culture have suggested substantial variation[16–18]. In Thailand, higher multiplication rates were reported for *P. falciparum* from severe malaria patients than from mild malaria patients[16], but a subsequent study in Uganda indicated only a marginal difference between severe and mild malaria[18], and samples from Mali or Kenya showed no difference between clinical groups[17]. However, such assays are subject to potentially confounding determinants, including effects on parasite viability by host inflammatory responses or sample handling prior to laboratory culture. More precise multiplication rate estimates can be obtained for laboratory-adapted parasite strains or clinical isolates after the first few weeks of culture *ex vivo*, using replicate exponential growth assays in different donor erythrocytes, which has demonstrated a wide range of multiplication rates among eleven West African *P. falciparum* clinical isolates tested[19]. Substantial variation in multiplication rates have also been shown among seven *P. falciparum* clones from Southeast Asian isolates[20], further indicating the likely importance of within-population variation. Multiplication rates of four of the West African isolates cultured for up to three months remained lower than in older laboratory adapted control strains[19], although adaptation might occur over a longer period as novel alleles take time to increase to high frequencies in culture[21].

To investigate naturally occurring parasite multiplication rate variation in depth, including potential changes during culture adaption, and the relationship with original *in vivo* parasitaemia in patients, a larger sample of new *P. falciparum* clinical isolates from a highly endemic area were tested and whole genome sequencing performed at different timepoints over five months of culture. This revealed significant variation in multiplication rates among the isolates, as well as a gradual increase in multiplication rates and loss of within-isolate genomic diversity over time in culture. New mutants only attained low to moderate frequencies in the cultures, indicating that most of the variation is natural and a large component of the change over time is due to parasite plasticity. Significantly, at all timepoints the multiplication rates were significantly positively correlated with levels of parasitaemia measured in the individual patients, indicating that intrinsic variation in parasite multiplication rate is a probable determinant of clinical infection intensity.

## Results

### *P. falciparum* multiplication rates increase over time in culture.
Twenty-four new clinical isolates were tested in exponential multiplication rate assays initially after 25 days of continuous laboratory culture, when parasites could be effectively assayed in triplicate with erythrocytes from three anonymous donors rather than in heterogeneous patient erythrocytes. At this time point, assays were successfully performed on 18 of the isolates (Table 1, and Supplementary Fig. S1), showing exponential multiplication rates (per 48 h corresponding to an average asexual cycle time in erythrocytes) of individual isolates that varied from 2.0-fold to 8.0-fold, with a mean of 4.3-fold (Fig. 1a, and Table 1).

Each of the isolates was continuously cultured for five months, with exponential multiplication rates being tested again after 77 days (successful assay of 23 isolates), and after 153 days (successful assay of 19 isolates) (Table 1, and Supplementary Fig. S1). Multiplication rates increased over time in culture, with a mean of 5.1-fold after 77 days, and 6.4-fold after 153 days of culture (Fig. 1a, and Table 1). The multiplication rates were significantly higher at the final time point compared with the first (Mann-Whitney test, $P = 0.0009$, Fig. 1a), having increased significantly over each of the time intervals ($P = 0.026$ for day 77 versus day 25, $P = 0.042$ for day 153 versus day 77, Fig. 1a). Of the 15 isolates successfully assayed at both the first and the last timepoints, all except one had a higher multiplication rate at the end (Wilcoxon signed rank test, $P = 0.0002$, Fig. 1b).

### Genome sequence diversity of parasites during culture adaptation.
To test whether the increasing multiplication rates during culture were due to selection of distinct genotypes or novel emerging mutants, we sequenced the parasite genomes from each of the culture timepoints at which multiplication rates were analysed (days 25, 77 and 153, genome accession numbers for all isolates are given in Supplementary Table S1). Notably, each isolate had a different genome sequence profile that was not closely related to any of the others at any of the timepoints (Fig. 2a). There was sufficient sequence read coverage to enable genome-wide analysis of variation (among 153,420 biallelic SNPs and 164,392 short indel polymorphisms mapped across the 14 haploid chromosomes) within 23 isolates at the first timepoint, 24 at the second, and 20 at the final timepoint. Some isolates retained the same distinct genomic profile throughout, while for others there was turnover of parasite genomic subpopulations over time (Fig. 2a).

We quantitatively analysed parasite genomic diversity within isolates using the $F_{WS}$ fixation index (isolates with values greater than 0.95 contain predominantly single genomes while lower $F_{WS}$ values indicate increasingly complex mixtures). At the first culture sample timepoint the mean $F_{WS}$ value was 0.70 (Fig. 2b), indicating a high mean level of parasite genomic diversity within isolates as seen in previous samples from malaria patients in the same area[22,23]. During the subsequent culture process the overall levels of within-isolate genomic diversity declined significantly over time, moving towards fixation with a mean $F_{WS}$ value of 0.83 on day 77, and 0.92 on day 153, (Fig. 2b, and Supplementary Table S2, Kruskal–Wallis test $P = 0.005$). Tracking individual isolates, those that contained single predominant genome sequences at the first timepoint (as indicated by high $F_{WS}$ values >0.95) maintained the same single genome profiles at subsequent timepoints (Fig. 2c, and Supplementary Table S2). In parallel, all of the isolates that initially contained mixed genome profiles lost some of their diversity during the culture period, as indicated by increasing $F_{WS}$ values between the first and last timepoint (Fig. 2c, Wilcoxon Signed rank test $P < 0.001$).

Isolates with single genome sequences at the first culture sample timepoint had higher multiplication rates than those with

**Table 1 *Plasmodium falciparum* multiplication rates in Ghanaian clinical isolates tested in exponential growth assays at different times over five months of continuous culture.**

| Patient isolate | Age (yrs) | Parasites μl$^{-1}$ blood | Hb level g dl$^{-1}$ | Multiplication rate (95% CI) per 48 h after: | | |
|---|---|---|---|---|---|---|
| | | | | Day 25 | Day 77 | Day 153 |
| 271 | 6 | 89,625 | 11.6 | 2.6 (2.1–3.2) | 6.6 (4.8–9.1) | 7.6 (5.5–10.6) |
| 272 | 2 | 47,987 | 9.0 | 4.9 (3.8–6.5) | 4.2 (3.3–5.5) | 5.2 (4.6–6.1) |
| 273 | 3 | 38,056 | 12.0 | 3.8 (2.5–5.6) | 3.3 (2.6–4.2) | 5.9 (4.7–7.2) |
| 274 | 8 | 25,499 | 11.1 | 4.4 (3.3–5.7) | 6.0 (4.4–8.3) | 5.7 (3.5–9.1) |
| 275 | 2 | 20,210 | 9.5 | 3.6 (2.9–4.6) | 3.6 (2.9–4.4) | 4.5 (3.7–5.6) |
| 276 | 7 | 5,985 | 8.8 | 5.6 (5.2–5.9) | 4.4 (3.6–5.3) | 5.2 (3.8–7.2) |
| 277 | 2 | 20,493 | 9.9 | 2.0 (1.7–2.4) | 4.4 (3.3–5.8) | —[b] |
| 278 | 3 | 92,225 | 8.4 | —[a] | 6.0 (4.8–7.9) | 10.5 (7.9–13.9) |
| 279 | 8 | 37,410 | 11.5 | —[a] | 2.8 (2.2–3.5) | —[a] |
| 280 | 9 | 64,896 | 12.5 | 5.6 (3.5–9.1) | 4.6 (3.6–5.6) | 7.5 (6.2–9.0) |
| 281 | 6 | 2,640 | 11.3 | —[a] | 5.8 (4.8–7.2) | —[c] |
| 282 | 5 | 20,776 | 9.5 | 3.1 (2.5–3.8) | 2.8 (2.2–3.5) | 4.7 (3.6–6.2) |
| 284 | 4 | 36,936 | 10.6 | 2.5 (2.3–2.7) | 3.8 (3.2–4.6) | 6.0 (5.1–7.1) |
| 285 | 9 | 11,700 | 13.3 | 3.2 (2.8–3.6) | 3.6 (3.0–4.4) | 5.6 (4.1–7.7) |
| 286 | 7 | 59,304 | 12.0 | 3.6 (3.3–4.0) | 6.9 (4.0–10.0) | 8.7 (6.7–11.3) |
| 287 | 5 | 74,784 | 12.5 | —[a] | 6.3 (4.8–7.9) | 4.0 (3.4–4.8) |
| 288 | 3 | 44,714 | 10.3 | —[a] | 4.2 (2.5–6.3) | 6.5 (5.1–8.3) |
| 289 | 9 | 42,512 | 12.4 | 3.8 (2.8–5.1) | 5.5 (4.2–7.6) | —[c] |
| 290 | 4 | 14,364 | 11.3 | —[a] | —[a] | 6.5 (5.4–7.8) |
| 291 | 10 | 15,745 | 9.3 | 2.4 (2.2–2.7) | 6.3 (4.4–9.6) | 4.4 (3.5–5.4) |
| 292 | 3 | 95,370 | 7.1 | 6.9 (5.5–8.7) | 5.5 (3.6–8.3) | 7.3 (5.4–9.9) |
| 293 | 13 | 38,931 | 13.3 | 6.0 (4.4–8.1) | 7.2 (5.0–11.0) | 9.9 (6.9–14.4) |
| 294 | 6 | 61,054 | 11.0 | 5.9 (4.3–8.2) | 6.9 (5.8–8.3) | 6.7 (5.4–8.3) |
| 296 | 2 | 108,931 | 8.4 | 8.0 (6.4–10.0) | 7.2 (4.0–10.0) | —[c] |

All exponential growth assay data for all biological replicates are plotted in full in Supplementary Fig. S1, and numerical data for all measurements are given in Supplementary Data File S1. Confidence intervals on the multiplication rates are derived by logistic regression fitting to exponential growth taking all experimental replicates into account for each isolate (all replicate datapoints shown in Supplementary Fig S1 and Supplementary Data File S1). Missing data for some isolates are indicated by superscripts in the table ([a]assay did not meet pre-determined replicate requirements for accuracy as outlined in Methods; [b]culture of one isolate was lost before the final timepoint; [c]contamination of three isolates at the final timepoint was detected so data were excluded). Genome sequence accession numbers, and genome-wide complexity measurements of isolates at all timepoints are given in Supplementary Tables S1 and S2.

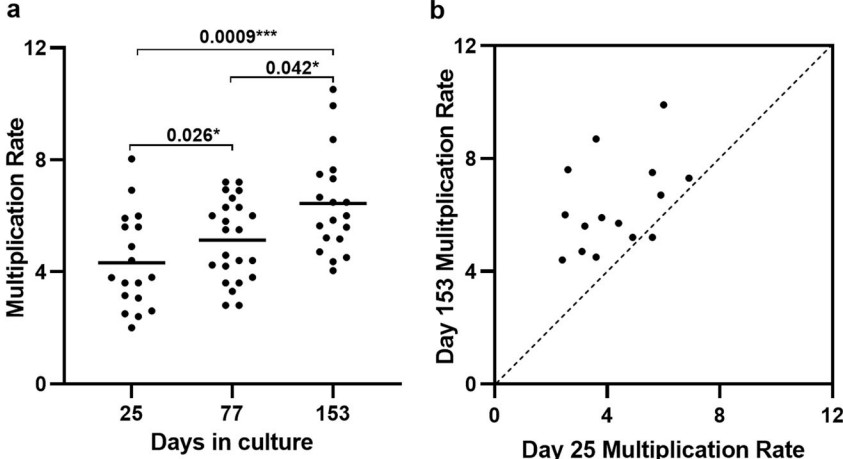

**Fig. 1 Parasite multiplication rates in a panel of 24 new Ghanaian *P. falciparum* clinical isolates over five months of continuous culture. a** Exponential multiplication rates per 48 h at the first assay timepoint (after 25 days) were successfully determined for 18 of the isolates, showing a mean multiplication rate of 4.3-fold, with a wide range among the different isolates. Multiplication rates of the clinical isolates increased over time, with a mean of 5.1-fold after 77 days (assay of 23 isolates) and 6.4-fold after 153 days (19 isolates) (Mann–Whitney test, $P = 0.0009$ for day 153 versus day 25, and $P = 0.015$ for day 77 versus day 25, $P = 0.042$ for day 153 versus day 77). The exponential multiplication plots with experimental replicates for all isolates are shown in Supplementary Fig. S1, all numerical data are given in Supplementary Data File S1, and 95% confidence intervals of the estimated multiplication rates based on logistic regression of the experimental replicates are given in Table 1. **b** All except one of the 15 isolates that were tested at both the first and last timepoints had an increased multiplication rate at the last timepoint (Wilcoxon rank sum test, $P = 0.0002$; dashed line of equivalence shown) although wide variation among the isolates remained.

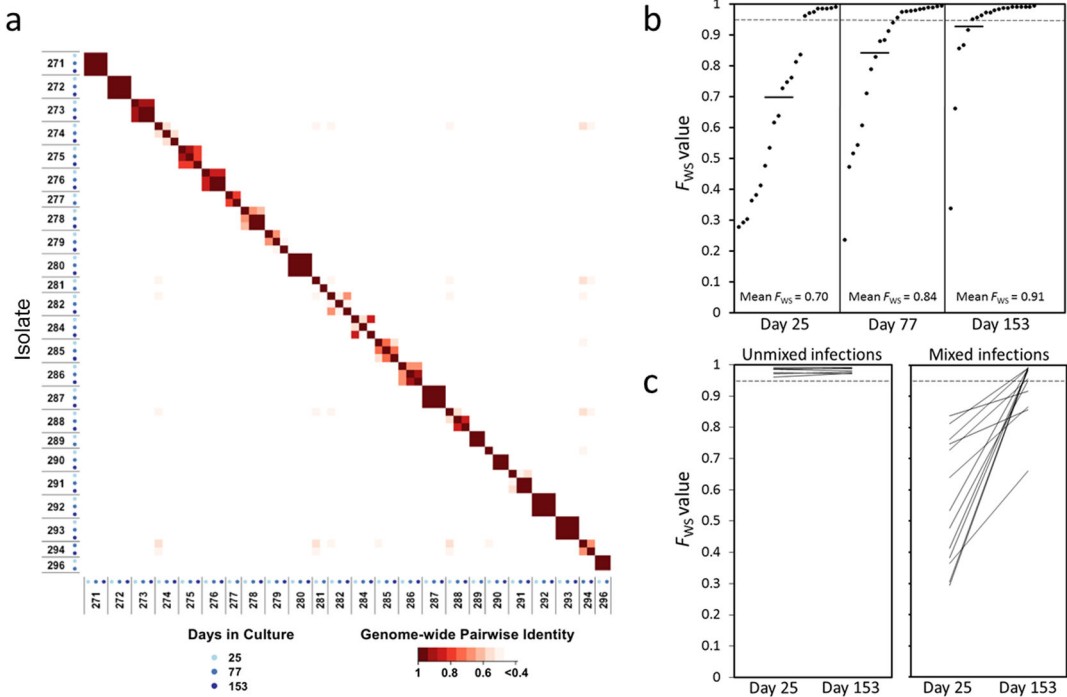

**Fig. 2 Parasite genome sequence profiles within *P. falciparum* clinical isolates analysed over time in culture. a** Matrix of pairwise comparisons indicates the degree of identity between genomic profiles of parasites at different times in culture. The identity index is calculated from the within-sample allele frequencies of 40,000 high-quality SNPs distributed throughout the 14 chromosomes of the parasite genome. Axes are labelled with the parasite isolate identifiers and the sequential cultured timepoints having different shaded blue dots. As expected, all isolates were different from each other throughout, and some had genomic profiles that remained the same over time while others showed some changes during culture. Sequence accession numbers for all sample timepoints are given in Supplementary Table S1, and genome-wide sequence coverage data are given in Supplementary Table S2. **b** Genomic diversity gradually reduced over time in isolates that initially had mixed genotypes. The within isolate genome-wide fixation index $F_{WS}$ is plotted for each timepoint. Low values of $F_{WS}$ indicate isolates with a high level of genomic diversity, and values closer to 1.0 indicate where there is little within-isolate diversity (the dotted line indicates $F_{WS} = 0.95$ as values above this indicate isolates with a predominant single genotype). Mean values are shown with horizontal bars (0.70 at day 25, 0.83 on day 77, 0.92 on day 153). The $F_{WS}$ values for all isolates at all timepoints are given in Supplementary Table S2. **c** Comparisons between the first and last timepoints for individual isolates show that single genotype isolates remained unmixed, while all isolates that were mixed at the beginning became less mixed (increased $F_{WS}$ values) by the end of the culture period.

multiple genome sequences (Mann–Whitney test, $P = 0.035$, Supplementary Fig. S2). Similarly, at the second culture timepoint, isolates with single genome sequences had higher multiplication rates ($P = 0.036$), but by the final timepoint when most isolates contained single predominant sequences there was no significant difference. Notably, the amount of change in multiplication rate during culture was not significantly correlated with the degree of change in parasite genomic diversity (delta $F_{WS}$) for individual isolates (Wilcoxon Signed rank test, $P > 0.05$ for each interval between the timepoints tested).

**Low frequency novel variants in culture including premature stop codon mutants.** We next performed a scan to detect emerging sequence variants (including potential mutants) during culture, focusing on six isolates that had single genome sequences at the beginning and that had sequence data for all of the timepoints (identifiable in Fig. 2a). Throughout the culture period, no SNP or indel variant that was absent at the beginning attained a frequency of 20% at any timepoint in any isolate, except for isolate 280 in which a premature stop codon nonsense mutant at codon position 285 in the *EPAC* gene on chromosome 14 (locus PF3D7_1417400) as well as a nonsynonymous mutant in a Clathrin gene on chromosome 12 (PF3D7_1219100) were detected in just over half of the reads at the final timepoint (Fig. 3a). A different premature stop codon in the same *EPAC* gene (at codon

position 2049) was seen in isolate 272 in approximately 10% of reads on day 77, but this was not detected at the final timepoint. The *EPAC* gene has previously been shown to contain stop codons in culture adapted parasites, but not in any uncultured parasites[21], and the mutants are predicted to lead to loss of function as they are upstream of the predicted catalytic domain (Fig. 3b). A premature stop codon mutant in an AP2 transcription factor gene on chromosome 13 (PF3D7_1342900) was seen in 16% of reads in isolate 271 at the final timepoint. Independent premature stop codon mutants in this gene have been reported to emerge in other clinical isolates during culture adaptation[21]. As the frequencies of mutants were low in the cultures here, with the exception of isolate 280 at the final timepoint, the overall multiplication rate phenotypes would not be significantly affected by their presence.

**Numbers of merozoites per schizont and counts of gametocytes in culture.** It was considered that the multiplication rate variation might be due to varying numbers of merozoites produced in schizonts. To explore this, counts of merozoites were performed on mature schizonts in stage-synchronised cultures of nine of the clinical isolates that had shown a range of multiplication rates, using chemical treatment to prevent merozoite egress from schizonts (Supplementary Fig. S3). Mean numbers of merozoites per schizont ranged from 16.5 (for isolate 273) to 24.2 (for isolate

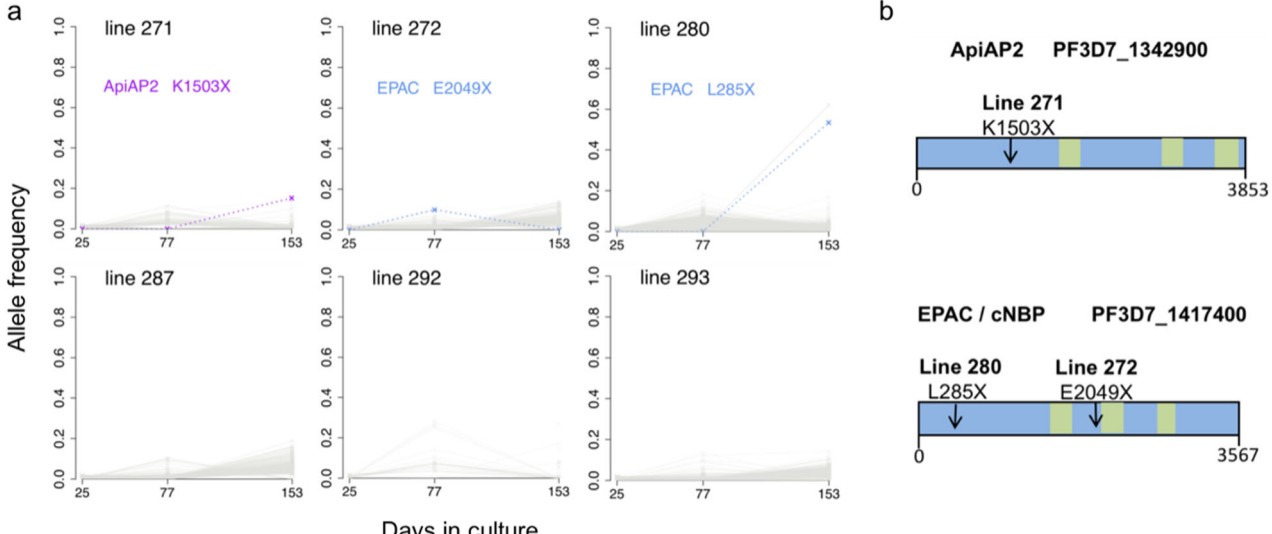

**Fig. 3 Selection of novel SNPs during culture adaptation of single-genotype *P. falciparum* clinical isolate lines. a** Allele frequencies at each sampled time point were determined by alternative sequence read counts for each variant SNP or short indel within each isolate. Novel alleles not detectable at the first time point are plotted, with coloured dashed lines representing nonsense mutants causing premature stop codons, and grey shading representing variant calls that do not affect the integrity of coding sequences. Isolate line 271 had a nonsense mutant in an *AP2* gene (locus *PF3D7_1342900*) detectable by the final timepoint. Line 272 had a nonsense mutant in the *EPAC* gene (locus *PF3D7_1417400*) detected at low frequency, while line 280 had another mutant in the *EPAC* gene detected at higher frequency by the final timepoint. **b** Gene models with an arrow indicating the mutation position in the *AP2* and *EPAC* genes for each emerging nonsense SNP identified in panel A. Green boxes indicate predicted catalytic and AP2 domains. The numbers of codons in each gene are indicated underneath each scheme (full information on each gene may be accessed on the PlasmoDB browser www.plasmodb.org[46]).

293). This range of ~ 1.5-fold is notable, although it is much lower than the multiplication rate variation at any timepoint, and correlation with multiplication rates of these isolates was not significant (Spearman's rho = 0.46, $P = 0.20$, Supplementary Fig. S3). While numbers of merozoites produced in schizonts could contribute to multiplication rate variation, these measurements indicate that they do not explain most of the variation, and that other cellular processes must be involved.

Differential counting of parasite stages was performed throughout the culture period, showing that a small minority of parasites were gametocytes. Gametocyte production in *P. falciparum* usually occurs at low rates that could have a minor effect on multiplication rates, so correlations were explored, and the proportions of gametocytes compared with all parasite stages observed during two-week periods surrounding the times of the multiplication rate assays are shown in Supplementary Table S3. For any isolate, the proportion of parasites that were gametocytes was never in excess of 10%, the mean across all isolates being 5% for the period corresponding to day 25, 2% for day 77, and 1% for day 153. Across all isolates, there was no significant correlation between proportions of gametocytes in the continuous culture and the multiplication rates in the exponential growth assays (non-significant trends showed varied direction among the timepoints: for day 25, Spearman's rho = −0.42, $P = 0.08$; for day 77, Spearman's rho = 0.16, $P = 0.47$; for day 153, Spearman's rho = 0.44, $P = 0.06$). It is therefore clear that the significant multiplication rate variation described here is not caused by gametocyte production.

***P. falciparum* multiplication rates correlate with levels of parasitaemia in patients.** The parasite multiplication rates in culture were significantly and consistently positively correlated with parasitaemia levels measured in the blood of patients at clinical presentation (Fig. 4, and Table 1). The correlation with patient parasitaemia that was clear in univariate analysis (Fig. 4a) was more significant in multivariate analysis that adjusted for

patient age and haemoglobin levels, in analysis of each of the culture time points tested separately ($P = 0.013$ on day 25, $P = 0.003$ on day 77, $P = 0.008$ on day 153) (Fig. 4b and Supplementary Table S4). The same multivariate analysis showed that multiplication rates were negatively but non-significantly correlated with patient haemoglobin levels, and slightly positively correlated with age (only marginally significant for one of the timepoints, Fig. 4b and Supplementary Table S4). This indicates that parasite multiplication rate contains a strong component of natural variation maintained over time in culture, and that this intrinsic parasite variation is a determinant of the density attained in the blood of patients.

## Discussion

These findings support a hypothesis that *P. falciparum* multiplication rate variation is an important virulence determinant, directly affecting the intensity of blood stage infections[4,16]. The significance of correlations between parasite multiplication rates in culture and peripheral blood parasitaemia in patients is notable, as effective parasite multiplication *in vivo* will also be limited by acquired immune responses[24,25], and total parasite load is sometimes only loosely correlated with peripheral blood parasitaemia due to sequestration of some developmental stages away from circulation[3]. Importantly, the wide variation in multiplication rates described here is not explainable by parasites having variable rates of switching from asexual replication to sexual differentiation for mosquito transmission. Almost all *P. falciparum* isolates show gametocyte conversion rates to be less than 5% per cycle, within Ghana and elsewhere[26], and there was no significant correlation between the multiplication rates and proportions of gametocytes at any timepoint in this study.

The mechanisms of multiplication rate variation remain to be determined, and might involve more than one phase of the replicative cycle. The mean numbers of merozoites within mature schizonts varied ~1.5 fold (from approximately 16 to 24) among isolates we measured. This range is insufficient to explain most of

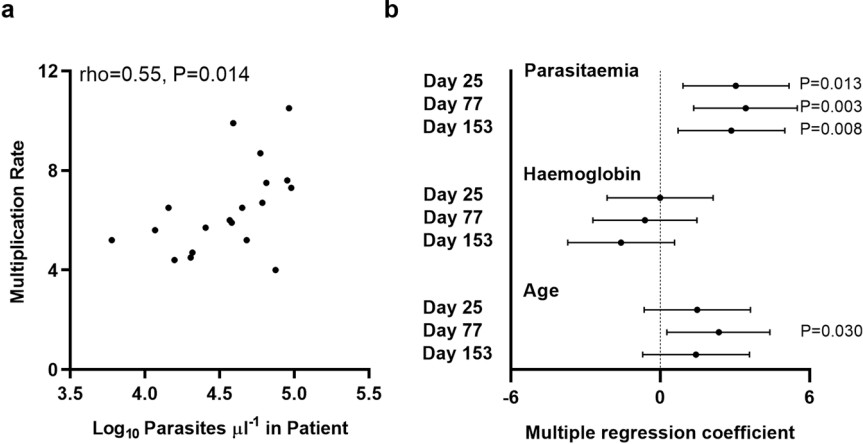

**Fig. 4 Parasite multiplication rates correlate significantly with levels of parasitaemia measured in patients. a** Univariate significant correlation with patient parasitaemia levels for multiplication rates assayed after 153 days of culture (earlier culture timepoints showed similar positive correlations). **b** Multiple regression analysis adjusting for patient age and haemoglobin levels shows a significant positive correlation between multiplication rates and patient parasitaemia at all culture timepoints. The coefficient on the x-axis is equivalent to the standard z-score with 95% confidence intervals. There were non-significant residual negative correlations with haemoglobin levels, and slight positive residual correlations with age. *P*-values are indicated for statistically significant correlations. Data for each of the variables in individual patients are given in Table 1, and the multiple regression coefficients are given in Supplementary Table S4.

the variation in multiplication rates, and no significant correlation was shown here, but it could plausibly contribute a minor component of the variation and deserves future attention although other phases of the cycle may have a greater effect. The typical multiplication cycle length is considered to be 48 h, and this period was used as the nominal generation time for the standardised analysis of multiplication here, but actual duration of multiplication cycles have been shown to vary among several cultured clones by up to 1.5 fold[27,28], so it would be interesting to investigate this parameter in clinical isolates. It is also possible that transcriptome analyses might identify genes that regulate mechanistic differences. By sampling multiple biological replicates of isolates to ensure robust comparisons, analysis of mature schizont transcriptomes has highlighted some genes with transcript levels that differ between long-term laboratory lines and clinical isolates, and other genes with levels that vary among individual clinical isolates[29], but larger numbers of isolates would be recommended in order to scan for phenotypic associations. Analyses of all phases of the multiplication cycle in large numbers of clinical isolates will be challenging, although it will be an important long-term target for scaled-up single-cell transcriptome approaches[30].

The observation that parasite multiplication rates did not correlate positively with levels of genomic diversity within isolates suggests that multiplication rates are not generally accelerated in co-infections with different genotypes, but does not entirely exclude a possibility that multiplication rates are modified by complex interactions *in vivo* within hosts having varied nutritional and immune status. This study focused on parasite multiplication at cultured timepoints when biological replicates could be effectively performed, so we analysed genome sequences at these timepoints, but the mean level of within-isolate genomic diversity at the first timepoint was similar to what we previously described for other isolates sampled in the same area and sequenced without culturing[22]. A hypothesis that malaria parasites facultatively increase their multiplication rate in response to the presence of competing parasites has received some support from studies of rodent malaria parasites within the blood[31], although these showed varying results. Experiments with *P. chabaudi* in mice have suggested that alterations of virulence occur during coinfections of different parasite clones[32], but no

such effects of coinfection were seen in experiments with *P. yoelii* in mice[33]. These species are distantly related to *P. falciparum* and other human malaria parasites so they are not predictive models. Nonetheless, studies of murine malaria have usefully highlighted the potential importance of parasite plasticity in determining virulence phenotypes[34], as well as the major effects that individual emerging mutants can have[35].

Comparative genomic data indicate that *P. falciparum* has adapted specifically to human hosts[36,37], having been originally derived as a zoonosis from gorillas[2], and it retains a high adaptive capacity. Notably, parasites in the first several months of culture after clinical isolation here were only minimally affected by mutants that are likely to become more common over longer periods of time. This supports prioritisation of analysis of parasite phenotypes within a limited period from original sample isolation, in order to minimise effects of culture adaptation mutants. Here, the malaria parasite multiplication rates show plasticity that cannot be explained by genomic changes, particularly in the isolates that each contained a single predominant genome sequence throughout. In an earlier study, significant changes in multiplication rates were not detected among four clinical isolates cultured for short extended periods of up to three months[19], as this was too few isolates for a thorough analysis of variation over time. Although the present study is much more informative, we note that further discoveries may be possible by analysing even larger numbers of isolates and sampling different endemic populations.

Although they did not explain the multiplication rate variation here, it is interesting to note that loss-of-function mutations in particular genes appear to be selectable and are a feature of long-term cultured parasites. Emerging mutants introducing premature stop codons in the *EPAC* gene (not essential for growth in culture although it encodes a protein predicted to have cyclic nucleotide binding sites)[38] were detected in two of the six single genotype isolates analysed, but only increased in frequency substantially in one isolate by the final timepoint. Loss-of-function *EPAC* mutants were previously described in several long-term adapted laboratory strains[21], but this is the first observation of emergence in cultured clinical isolates. As seen in one of the isolates here, loss-of-function mutants of the transcription factor *ApiAP2* gene PF3D7_1342900 were previously observed to

emerge during culture of Gambian clinical isolates[21], although the functional consequence of these is yet to be determined. Two other *ApiAP2* genes including *AP2-G* that is important for gametocytogenesis have loss-of-function mutants detected in multiple long-term cultured laboratory clones[21], so we would predict novel mutants in these genes may eventually be seen to emerge within clinical isolates after more extended sampling and continuous culturing for very long periods.

This study clearly indicates intrinsic variation in *P. falciparum* multiplication rates to be a determinant of natural infection intensity. This should encourage efforts to identify disease risk factors by association with parasite genome-wide variation, although the observed parasite plasticity indicates that attention is also needed to epigenetic regulation of transcriptional variation[39], and further experimental analyses will be required to understand mechanisms of varying multiplication rates. Finally, adaptive multiplication rate variation might enable parasite maintenance at low densities in asymptomatic chronic infections, which would adversely affect prospects for elimination or eventual eradication of malaria[40].

## Methods

**Parasite sampling from malaria patients.** Blood samples were collected from clinical malaria cases attending Ghana government health facilities at Navrongo (located 10°53′5″N, 1°5′25″W) in Kassena-Nankana East Municipality, in the Upper East Region of northern Ghana in 2011. Navrongo is in the Sudan Savanah vegetational zone in West Africa, with hyperendemic malaria transmission dependent upon seasonal rainfall, and entomological inoculation rates were previously estimated to be a few hundred infective bites per person per year[41]. Patients were eligible for recruitment into the study if they were aged 2–14 years and had uncomplicated clinical malaria, testing positive for *P. falciparum* malaria by Rapid Diagnostic Test (First Response®, Transnational Technologies, UK) and microscopical examination of a Giemsa stained thick blood smear. Venous blood samples (up to 5 ml) were collected into heparinised vacutainer tubes (BD Biosciences, CA, USA), and a small volume was used for measurement of haemoglobin concentration and peripheral blood parasitaemia, the latter by counting the number of *P. falciparum* parasites per 200 leukocytes on a Giemsa stained thick blood smear and multiplying by the total leukocyte count obtained by automated haematology analysis (all blood samples analysed in the present study had *P. falciparum* alone except for isolate 290 that also contained *P. malariae*, a species that does not grow in continuous culture). Half of the remaining blood sample volume was cryopreserved in glycerolyte at −80 °C, and the other half was processed to remove leukocytes before extraction of DNA from parasites in erythrocytes for whole genome sequencing. Written informed consent was obtained from parents or other legal guardians of all participating children, and additional assent was received from the children themselves if they were 10 years or older. Antimalarial treatment and other supportive care was provided according to the Ghana Health Service guidelines. Approval for the study was granted by the Ethics committees of the Ghana Health Service, the Noguchi Memorial Institute for Medical Research at the University of Ghana, the Navrongo Health Research Centre and the London School of Hygiene and Tropical Medicine.

**Parasite culture.** Cryopreserved patient blood samples were transferred by shipment on dry ice to the London School of Hygiene and Tropical Medicine where culture was performed. Samples were thawed from glycerolyte cryopreservation and *P. falciparum* parasites were cultured continuously at 37 °C using standard methods[42] as follows (no isolates were pre-cultured before the thawing of cryopreserved blood in the laboratory on day 0 of the study). The average original volume of cells in glycerolyte in each thawed vial was approximately 1 ml, which yielded an erythrocyte pellet of at least 250 µl in all cases. Briefly, 12% NaCl (0.5 times the original volume) was added dropwise to the sample while shaking the tube gently. This was left to stand for 5 mins, then 10 times the original volume of 1.6% NaCl was added dropwise to the sample, shaking the tube gently. After centrifugation for 5 min at 500 *g*, the supernatant was removed and cells were resuspended in the same volume of RPMI 1640 medium containing 0.5% Albumax™ II (Thermo Fisher Scientific, Paisley, United Kingdom). Cells were centrifuged again, supernatant removed and the pellet resuspended at 3% haematocrit in RPMI 1640 medium supplemented with 0.5% Albumax II, under an atmosphere of 5% $O_2$, 5% $CO_2$, and 90% $N_2$, with orbital shaking of flasks at 50 revolutions per minute. Replacement of the patients' erythrocytes in the cultures was achieved by dilution with fresh erythrocytes from anonymous donors every few days, so that after a few weeks of culture parasites were growing virtually exclusively in erythrocytes from new donors. All clinical isolates were cultured in parallel in separate flasks at the same time, so that the donor erythrocyte sources were the same for all the different isolates, enabling comparisons without confounding from heterogeneous erythrocytes.

**Parasite multiplication rate assays.** Exponential multiplication rate assays were performed using a method previously described[19], summarised briefly as follows. Prior to each of the assays which were initiated after 25, 77 and 153 days of continuous culture of the isolates, fresh blood was drawn from several anonymous donors (anonymous volunteer staff at the London School of Hygiene and Tropical Medicine who had not had malaria or recently taken any antimalarial drugs, and who did not have any known haemoglobin variants), and erythrocytes were stored at 4 °C for no more than 2 days then washed immediately before use. Each of the assays for each isolate was performed in triplicate, with erythrocytes from three different donors in separate flasks. Asynchronous parasite cultures were diluted to 0.02% parasitaemia at the start of each assay which was conducted over six days. Every 48 h (day 0, 2, 4, and 6) a 300 µl sample of suspended culture was taken (100 µl pelleted for Giemsa smear, 200 µl for DNA extraction and qPCR), and culture media were replaced at these times.

Following extraction of DNA from the assay points, qPCR to measure numbers of parasite genome copies was performed using a previously described protocol targeting a highly conserved locus in the *P. falciparum* genome (the *Pfs25* gene, PF3D7_1031000)[19]. Analysis of the parasite genome numbers at days 0, 2, 4 and 6 of each assay was performed by qPCR, and quality control was performed to exclude any assays with less than 1000 parasite genome copy numbers measured at the end of the assay. Quality control also removed any single points where a measurement was either lower or more than 20-fold higher than that from the same well two days earlier in the assay, and any outlying points among the biological triplicates that had a greater than two-fold difference from the other replicates on the same day. Assays were retained in the final analysis if there were duplicate or triplicate biological replicate measurements remaining after the quality control steps, and if they showed a coefficient of determination of >0.90 for the multiplication rate estimates using all data points from assay between days 0–4 or days 0–6 (whichever $r^2$ was greater indicating higher accuracy). For each assay, an overall parasite multiplication rate (defined as per 48-h typical replicative cycle time) was calculated with 95% confidence intervals using a standard linear model with GraphPad PRISM. The long-term laboratory adapted *P. falciparum* clone 3D7 was assayed in parallel as a control in all assays, consistently showing a multiplication rate of approximately 8.0 fold per 48 h as described previously[19].

**Measuring numbers of merozoites per mature schizont.** Parasites were isolated by magnetic purification using magnetic LD Separation (MACS) columns (Miltenyi Biotech). Each column was washed twice in 3 ml of culture medium at room temperature. Parasite culture erythrocytes were pelleted by centrifugation at 500 *g* for 5 min, then re-suspended in 3 ml culture medium per 1 ml of packed cell volume. The re-suspended cells were bound to the MACS column, which was then washed three times with 3 ml culture medium, and schizonts were eluted twice by removing the magnet from the column and forcing 2 ml culture medium through the column into a 15 ml collection tube. Finally, the schizonts were pelleted by centrifugation at 500 *g* for 5 min and re-suspended in culture medium with 10 µM E64 in a 12-well plate and incubated for 4 h at 37 °C. Giemsa stained thin blood films were prepared for examination after this period, thus ensuring the majority of schizonts were fully segmented for counting. A minimum of 50 schizonts were counted for each replicate.

**Genome sequencing of parasites at different timepoints in culture.** DNA extracted from parasites at each of the assayed culture timepoints was used for whole-genome Illumina short-read sequencing. Library preparation, sequencing and quality control was performed following internal protocols at the Wellcome Sanger Institute, similarly to previous sequence generation from clinical isolates. Genetic variants were called using a pipeline developed by the MalariaGEN consortium (ftp://ngs.sanger.ac.uk/production/malaria/Resource/28). Briefly, short reads were mapped with the BWA algorithm to the *P. falciparum* 3D7 reference genome sequence version 3. Single Nucleotide Polymorphisms (SNPs) and short insertions-deletions (indels) were called using GATK's Best Practices. Variants with a VQSLOD score <0 or outside the core genome (such as the *var*, *rif* and *stevor* gene families) were excluded, resulting in 153,420 biallelic SNPs and 164,392 short indels.

**Within-isolate genomic diversity.** Estimation of parasite diversity within each isolate relative to overall local population diversity was performed using the $F_{WS}$ index[43,44]. Allele frequencies were calculated at every SNP position for each isolate individually, with *p* and *q* representing the proportions of read counts for the minor and major alleles. All SNPs were then assigned to ten minor allele frequency (MAF) intervals representing the proportional frequency of the minor allele at each SNP across the population, with the ten equally sized intervals ranging from 0–5% up to 45–50%. Levels of within-host ($H_w$) and local parasite subpopulation ($H_s$) heterozygosity for each SNP were calculated as $H_w = 2*p_w*q_w$ and $H_s = 2*p_s*q_s$. The mean $H_w$ and $H_s$ of each MAF interval were then computed from the corresponding heterozygosity scores of all SNPs within that particular interval. The resulting plot of $H_w$ against $H_s$ for each isolate was produced and a linear regression model was used to determine a value for the gradient $H_w/H_s$, with $F_{WS} = 1 - (H_w/H_s)$. All $F_{WS}$ analyses were performed using custom scripts in R.

**Emergence of new mutants**. Analysis focused on isolates that each contained a predominant single genome at the beginning (based on $F_{WS}$ values of >0.95), and had data for all timepoints, following methods broadly similar to those used in analysis of Gambian clinical isolates previously[21]. Allele frequencies of intragenic SNPs and frameshift-causing indels covered by at least 10 reads were plotted for each timepoint to scan for the emergence of new mutant alleles. The quality of the mapped sequence reads for each of these cases were inspected visually using the Savant software[45].

**Statistics and reproducibility**. Statistical comparisons of multiplication rates between independent categorical groups was performed using the Mann–Whitney test, whereas comparisons of paired sample distributions were performed using the Wilcoxon signed-rank test. Univariate correlations were assessed using Spearman's rank test with the rho coefficient. Multivariate analyses adjusting for covariates were performed using multiple regression analysis, with residual coefficients equivalent to z-scores being expressed with 95% confidence intervals to indicate deviation from null expectations. Any additional statistics or parameters including those relating to the genome sequence analysis are given in the relevant sections. All multiplication rate assays were performed using biological triplicate cultures, with all replicate data plotted for each point in the exponential growth, and used to estimate confidence intervals of the multiplication rate for each isolate. All qPCR assays of each biological replicate were performed in technical duplicate, and the mean used for estimating the parasite genome copy numbers at each point in each assay. Counts of numbers of merozoites per schizont were performed on preparations from two biological replicate cultures of each isolate and pooled for estimating means.

## Data availability

All genome sequence data have been deposited in the European Nucleotide Archive (76 accession numbers listed in Supplementary Table S1), and all SNP and indel genotypic calls are made available ftp://ngs.sanger.ac.uk/production/malaria/Resource/28. Supplementary Data File S1 gives all numerical data on all experimental replicates of the exponential multiplication rate assays that are plotted in full in Supplementary Fig. S1. Any other source data for figures and supplementary figures are available on request from the corresponding author.

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

## Acknowledgements

This study was supported by an Advanced Grant from the European Research Council (AdG-2011-294428), and funding from the Royal Society (AA110050) and the UK Medical Research Council (MR/S009760/1). Genome sequencing was coordinated by the MalariaGEN Resource Centre with funding from The Wellcome Trust (Grants 098051, 206194, 090770), and we thank Kim Johnson as well as the staff of the Sample Logistics, Sequencing, and Informatics facilities of the Wellcome Sanger Institute for their contribution to this process. We are grateful to all patients who participated in the study, and to staff of the Navrongo Health Research Centre for support with initial sample collection and storage.

## Author contributions

L.B.S. performed a large proportion of the experimental work and statistical analysis, and contributed significantly to design and interpretation. O.D.-I. performed a substantial proportion of the experimental work. A.C. performed bioinformatic analysis and contributed significantly to interpretation. J.A. contributed to the sample collection and experimental work. R.D.P. performed and advised on bioinformatic processing. S.G. performed genome sequencing and sequence data curation. E.D. performed genome sequencing and sequence data curation. D.P.K. supervised and resourced genome sequencing and sequence data management. G.A.A. contributed to study design, supervision and interpretation. D.J.C. led the design of the study, supervised the experimental work and data analysis, and led the writing with input from all authors.

## Competing interests

The authors declare no competing interests.
