## [Peer Review File · Communications Biology]

Reviewers' comments:

Reviewer #1 (Remarks to the Author):

The presented manuscript is well written and in line with previously reported data by the same group. The investigators propose that parasite multiplication rate is a determinant of *Plasmodium falciparum* virulence, intrinsically related to parasitemia levels of the patient at the time of parasite collection. To test this hypothesis the authors examined the multiplication rates in vitro of 24 isolates from Ghana over a period of ~5 months. The 3D7 clone, previously characterized by Murray et al, 2017, was used as control with a multiplication rate of ~8 fold/48 hours. Results suggest in vitro parasitemia levels associate with levels measured in vivo when each isolate was obtained. It also shows that isolate multiplication rates increase over time, a contrasting finding from the 2017 report by the same group. Finally, the work would benefit of further phenotypic characterization, including variation in merozoite numbers/schizont, erythrocyte invasion rate, knobs formation, and perhaps cell rosetting. Such information would provide further insights regarding the biological mechanisms of parasite growth and virulence proposed here.

Questions and suggestions:

Please define virulence and intensity of infection. This will help to give the readers a better picture of this biological aspect of malaria and frame the work presented.

Can other measurements such as number of merozoites/schizonts, erythrocyte invasion rate, knobs formation, and/or rosetting be obtained? Any of these further characterizations and comparisons between the 1st and last time points can provide new insights to the question of multiplication and virulence. It would be interesting to show if there is an association between multiplication rate and number of merozoites/schizont or with capacity to invade erythrocytes. Knobs formation can easily be measured by gelatin flotation and it is known to reduce over time in culture.

Is reference 7 appropriate when citing the neurosyphilis treatments from 1920-1950s?

The higher multiplication rates of isolates at the end of the experiment suggest the selection of a subpopulation of parasites better equipped to grow under these conditions. If this is the case, please explain the different findings by Murray et al 2017: "Four of the clinical isolates cultured for longer periods were assayed at three different time points after culture initiation (up to 76 or 100 days for each isolate). These showed no significant changes in multiplication rate over time in culture."

From results: "Twenty-four new clinical isolates were tested in exponential multiplication rate assays after different lengths of time of continuous laboratory culture, when parasites could be effectively tested in triplicate with erythrocytes from three different donors."

Please clarify and provide detailed information of previous time of continuous culture for each isolate. How does time to become "testable" relate to multiplication rate and is there information on the genotype of the sample directly obtained from the patient?

The authors suggest that the variability in asexual multiplication rates observed here cannot be explained by the rates of switching from asexual replication to sexual differentiation and cite reference 26. Were gametocytes observed in the patient sample? Please note if gametocytes were ever observed during the in vitro cultivation. Discuss how parasite multiplication rate, virulence, and transmission may be related.

Methods: was PCR for other human malarias performed? Could the presence of other malarias affect the initial adaptation?

Trager W and Jensen JB on 1976, cited by the authors in the methods, does not describe how to thaw parasites cryopreserved with glycerolyte. Considering this work is highly dependent on the

cultivation methods and the fact that currently most laboratories use modified conditions from the 1976 publication, I think it is important to describe a few further details: the size of cryopreserved samples, if the cryopreserved material came directly from patients or from short-term cultivations, steps to thaw, etc...

The group uses 3 different erythrocyte donors during the experiments, was haemoglobin genotyped? Is there information regarding alpha thalassemia prevalence among the donors? These can impact in vitro growth and therefore should be addressed.

"The long-term laboratory adapted *P. falciparum* clone 3D7 was assayed in parallel as a control in all assays, consistently showing a multiplication rate of approximately 8.0 fold per 48 hours as described previously 19" Should this be expected? Can't the multiplication rate increase over time for clones as well?

Reviewer #2 (Remarks to the Author):

It is known that *Plasmodium falciparum* disease severity is correlated with the number of blood stage parasites. The authors explore the relationship between intrinsic multiplication rate variation of *P. falciparum* in laboratory cultures, and the original in vivo parasitemia and clinical presentation for patient samples from a highly endemic area in West Africa. Previous studies measuring the relationship between multiplication rates and patient disease severity have yielded conflicting and inconclusive results. The current study is carefully designed to avoid both host response and sample handling variables; and parasite isolates were allowed to grow in culture for a period of weeks during which time multiplication rates were periodically measured. Both technical and biological replicates are included, with quantitation over multiple cycles and assay replicates using unrelated donor erythrocytes. The methodology used in this study has been previously published by this group. The current study differs from the previous one in that they link in vitro replication rates back to parasite levels in patients at clinical presentation. The authors found that not only are intrinsic multiplication rates inherently stable in laboratory culture as they had shown previously, but that they were significantly and consistently positively correlated with blood parasitemia levels measured in vivo. This is an interesting observation that adds to our understanding of the parasites' contribution to the virulence phenotype. The carefully controlled experiments provide confidence that the observations are robust and likely broadly applicable to other isolates and to our understanding of virulence in *P. falciparum*. The statistical analyses are appropriate and the work is well described and reproducible.

1. Laboratory culture conditions are not nutrient-limited and thus one would not expect to see alterations in replication rates due to competition between co-cultured strains. However, the nutrient environment in vivo is likely to differ considerably from laboratory conditions. Would you expect multiplication rates to be altered in response to competing parasites in the host?

2. What explanation can be given for mean multiplication rates increases over time for in vitro parasite cultures? Adaptation to culture conditions? What are the implications for interpreting biological significance of multiplication rates in long term cultures? It seems as though a very narrow window of time exists within which parasite phenotypes can be assessed in culture.

3. Why might loss of function mutations emerge repeatedly in culture?

Reviewer #3 (Remarks to the Author):

This study analyzes clinical isolates from pediatric uncomplicated malaria infections in Ghana in continuous culture, showing that multiplication rates are associated with the initial parasite levels in patients. The authors study the genomic diversity within clinical isolates and present novel findings examining the role of genomic diversity and parasite growth in culture. Multiplication rates increased over time in culture, but were lower than rates of long-term culture-adapted strains, as has been previously shown. Interestingly, within-isolate genomic diversity decreased over time. Novel mutations did not appear to affect multiplication rates. At the first two time points, isolates with a single dominant genome sequence appeared to have a higher multiplication rate than isolates with multiple genome sequences.

This analysis provides insight into the role of genomics in parasite multiplication rates and suggests the need for further exploration of determinants of multiplication rates. It represents a novel investigation into a fundamental characteristic of malaria parasites. The reasoning is straightforward, and the experiments are presented in a clear, well-written fashion. The statistical analyses appear valid, and the approach reproducible.

MAJOR COMMENTS:

The use of the word "virulent" causes some confusion. Here, "virulent" appears to be used to indicate high parasitemia levels in patients. This is confusing, given that all isolates are from children with uncomplicated malaria. No isolates from severe malaria infections are used, which is a more typical context of mention of a "virulent parasite". I suggest omitting the word "virulent" from the title.

The Discussion mentions epigenetic regulation of transcriptional variation as a promising area of further investigation. Discussion of potential transcriptional analyses that could provide insight into multiplication rate differences would be helpful here as well. Are such analyses of RNA from these isolates underway?

It would be beneficial to see a graph of multiplication rates for isolates with single genomes versus multiple genomes at each time point, ideally with the ability to see how rates for a given isolate tracks across time. The finding that genome diversity is not associated with multiplication rate at the third time point is interesting, and this graph would help to elucidate these associations over time.

MINOR COMMENT:

Fig 3: A point of confusion: the text states that isolate 280 has a nonsense mutation at codon position 1422, whereas the figure states L285X.

Responses to Reviewers

We appreciate these highly informed and constructive reviews of the manuscript. We have made changes to revise accordingly, as noted and explained in bold for clarity below the individual comments.

In addition to the editing of text correspondingly, we have added two new Supplementary Figures, one new Supplementary Table (shown after the responses below), and four new references. Following editorial instructions, we specify the line numbers where changes and additions have been made. Note - the line numbers refer to those in the 'track changes' marked up version of the manuscript, sent as a separate pdf supplementary file to facilitate visible review – therefore the line numbers given do not correspond to those in the 'clean' revised manuscript when 'track changes' are not in view.

Reviewer #1 (Remarks to the Author):

The presented manuscript is well written and in line with previously reported data by the same group. The investigators propose that parasite multiplication rate is a determinant of Plasmodium falciparum virulence, intrinsically related to parasitemia levels of the patient at the time of parasite collection. To test this hypothesis the authors examined the multiplication rates in vitro of 24 isolates from Ghana over a period of ~5 months. The 3D7 clone, previously characterized by Murray et al, 2017, was used as control with a multiplication rate of ~8 fold/48 hours. Results suggest in vitro parasitemia levels associate with levels measured in vivo when each isolate was obtained. It also shows that isolate multiplication rates increase over time, a contrasting finding from the 2017 report by the same group. Finally, the work would benefit of further phenotypic characterization, including variation in merozoite numbers/schizont, erythrocyte invasion rate, knobs formation, and perhaps cell rosetting. Such information would provide further insights regarding the biological mechanisms of parasite growth and virulence proposed here.

Questions and suggestions:

Please define virulence and intensity of infection. This will help to give the readers a better picture of this biological aspect of malaria and frame the work presented.

Authors note: We agree with this, and have clarified that we are referring to intensity of infection and not any other potential virulence phenotype (lines 30, 41-42, 229, 324, referring to the 'Track changes' visible manuscript shown as a separate pdf as a supplementary file). We realise that by having placing a lot of the actual details of our measurements in the Supplementary Information we had accidentally sacrificed some clarity regarding the frame of what was done. While much needs to remain in Supplementary Information due to limited space, we now see that the details in the first table are really more suited as a main Table in the paper (we initially prepared the table for this purpose but at the point of original submission we opted for a more highly succinct and streamlined presentation), so we have reinstated this which should be more useful to readers (lines 639-647). We have altered the title to remove the word 'virulence' (lines 1-2), also in response to the suggestion of Reviewer 3.

Can other measurements such as number of merozoites/schizonts, erythrocyte invasion rate, knobs formation, and/or rosetting be obtained? Any of these further characterizations and comparisons between the 1st and last time points can provide new insights to the question of multiplication and virulence. It would be interesting to show if there is an association between multiplication rate and

number of merozoites/schizont or with capacity to invade erythrocytes. Knobs formation can easily be measured by gelatin flotation and it is known to reduce over time in culture.

Authors note: We agree that, given these findings, there are many components of parasite phenotypic variation at a cellular level that it would be interesting and relevant to study, as potentially involved in affecting the rate of multiplication of recently isolated parasites from clinical infections. We have extended the Discussion (lines 242-260) to include suggestions of the reviewer, and hope that many investigators will contribute to this important field, which is beyond the scope of any one research group. We have generated data on numbers of merozoites/schizont in counts on fully mature schizonts blocked from egress, and now include these in the Results (lines 182-194) and Methods (lines 422-434), with an additional supplementary figure (Supplementary Figure S3). We have also included counts of gametocytes (lines 196-209), with an additional supplementary table (Supplementary Table S3).

Is reference 7 appropriate when citing the neurosyphilis treatments from 1920-1950s?

Authors note: We are grateful for this query – the reference was accidentally misplaced, and has been moved to the following sentence in the Introduction (line 62).

The higher multiplication rates of isolates at the end of the experiment suggest the selection of a subpopulation of parasites better equipped to grow under these conditions. If this is the case, please explain the different findings by Murray et al 2017: “Four of the clinical isolates cultured for longer periods were assayed at three different time points after culture initiation (up to 76 or 100 days for each isolate). These showed no significant changes in multiplication rate over time in culture.”

Authors note: We have now added reference to this in the Discussion (lines 290-295). The study by Murray et al. 2017 was not well powered to see changes over time, as only four isolates were cultured for periods beyond the first month, whereas the current study of 24 isolates cultured over 5 months had sufficient data to be able to statistically detect changes.

From results: “Twenty-four new clinical isolates were tested in exponential multiplication rate assays after different lengths of time of continuous laboratory culture, when parasites could be effectively tested in triplicate with erythrocytes from three different donors.”

Please clarify and provide detailed information of previous time of continuous culture for each isolate. How does time to become “testable” relate to multiplication rate and is there information on the genotype of the sample directly obtained from the patient?

Authors note: We have now made this clearer (lines 100-103, 357, 361-362). All isolates were cultured in parallel at the same time, using exactly the same conditions. No isolates were pre-cultured before thawing in the laboratory on day 0, as the cryopreserved blood samples were directly from patients. As we intentionally did not investigate the phenotypes until day 25 (by which time all isolates were growing in the same donor cells which avoided possible confounding variation that could have occurred if measurements were done earlier due to heterogeneous patient erythrocytes) we did not systematically analyse parasite sequences before day 25, although the average within-isolate diversity was similar to what we had previously seen when sequencing samples directly from patients (lines 265-269).

The authors suggest that the variability in asexual multiplication rates observed here cannot be explained by the rates of switching from asexual replication to sexual differentiation and cite reference 26. Were gametocytes observed in the patient sample? Please note if gametocytes were

ever observed during the in vitro cultivation. Discuss how parasite multiplication rate, virulence, and transmission may be related.

Authors note: We appreciate the query, and have now presented data on gametocyte counts during the culture of each of the individual isolates, focusing on two-week windows of time centred on the timepoints at which the cultures were sampled for the exponential growth rate assays. We have summarised these results in the Results text (lines 196-209), and added a new supplementary table (Supplementary Table S3). Consistent with the previous interpretation, the proportions of gametocytes were low in all cultures, and could not explain the substantial variation in multiplication rates. Although the interpretation has not changed, we have referred to this in the Discussion (lines 237-240).

Methods: was PCR for other human malaras performed? Could the presence of other malaras affect the initial adaptation?

Authors note: PCR was not performed for other human malaria parasite species, as these do not grow in continuous culture using under the methods here, so would not be present in culture at any of the times of assay. The other species are usually only present in a minority of infections in this area, at low levels in the blood. We now specify that microscopy identified one of the patient samples (number 290) to have had *P. malariae* in the blood along with *P. falciparum* at time of collection (lines 344-346), but no isolate had this or any other species identified alongside *P. falciparum* during culture.

Trager W and Jensen JB on 1976, cited by the authors in the methods, does not describe how to thaw parasites cryopreserved with glycerolyte. Considering this work is highly dependent on the cultivation methods and the fact that currently most laboratories use modified conditions from the 1976 publication, I think it is important to describe a few further details: the size of cryopreserved samples, if the cryopreserved material came directly from patients or from short-term cultivations, steps to thaw, etc...

Authors note: We agree that it is appropriate to give these details, and have added these now to the Materials and Methods (lines 362-373).

The group uses 3 different erythrocyte donors during the experiments, was haemoglobin genotyped? Is there information regarding alpha thalassemia prevalence among the donors? These can impact in vitro growth and therefore should be addressed.

Authors note: Theoretically if such erythrocyte variants were present in cultures they could influence growth, but all the erythrocyte donors for the assays were laboratory staff of LSHTM in London among which the general frequency of haemoglobin variants or thalassaemia is very low (lines 383-386), and all assays done in cells from triplicate donors included the long-term laboratory parasite clone 3D7 as a control, which consistently gave the multiplication rate of approximately 8-fold per 48 hours (lines 411-413).

“The long-term laboratory adapted *P. falciparum* clone 3D7 was assayed in parallel as a control in all assays, consistently showing a multiplication rate of approximately 8.0 fold per 48 hours as described previously 19” Should this be expected? Can’t the multiplication rate increase over time

for clones as well?

Authors note: We see a stable multiplication rate of clone 3D7 over time as described in this manuscript and also in the earlier work of Murray et al. 2017. We cannot exclude the possibility that some long-term clones could vary in multiplication rates, but we have seen consistent rates for long-term laboratory clones in exponential growth assays (with clone Dd2 being approximately 10-fold per 48 hours, while HB3 and D10 are approximately 8-fold, as described by Murray et al. 2017).

Reviewer #2 (Remarks to the Author):

It is known that Plasmodium falciparum disease severity is correlated with the number of blood stage parasites. The authors explore the relationship between intrinsic multiplication rate variation of P. falciparum in laboratory cultures, and the original in vivo parasitemia and clinical presentation for patient samples from a highly endemic area in West Africa. Previous studies measuring the relationship between multiplication rates and patient disease severity have yielded conflicting and inconclusive results. The current study is carefully designed to avoid both host response and sample handling variables; and parasite isolates were allowed to grow in culture for a period of weeks during which time multiplication rates were periodically measured. Both technical and biological replicates are included, with quantitation over multiple cycles and assay replicates using unrelated donor erythrocytes. The methodology used in this study has been previously published by this group. The current study differs from the previous one in that they link in vitro replication rates back to parasite levels in patients at clinical presentation. The authors found that not only are intrinsic multiplication rates inherently stable in laboratory culture as they had shown previously, but that they were significantly and consistently positively correlated with blood parasitemia levels measured in vivo. This is an interesting observation that adds to our understanding of the parasites' contribution to the virulence phenotype. The carefully controlled experiments provide confidence that the observations are robust and likely broadly applicable to other isolates and to our understanding of virulence in P. falciparum. The statistical analyses are appropriate and the work is well described and reproducible.

1. Laboratory culture conditions are not nutrient-limited and thus one would not expect to see alterations in replication rates due to competition between co-cultured strains. However, the nutrient environment in vivo is likely to differ considerably from laboratory conditions. Would you expect multiplication rates to be altered in response to competing parasites in the host?

Authors note: Potential interactions between parasites within the host may be more complex than in the artificial culture system, either due to varying nutrient environment or other host-related variables, so we agree that there may be interactions in vivo that could lead to modifications of multiplication rates and have added to the Discussion to note this (lines 262-265).

2. What explanation can be given for mean multiplication rates increases over time for in vitro parasite cultures? Adaptation to culture conditions? What are the implications for interpreting biological significance of multiplication rates in long term cultures? It seems as though a very narrow window of time exists within which parasite phenotypes can be assessed in culture.

Authors note: These are good questions which may be commented on while they remain open to some extent, so we have added to the Discussion accordingly (lines 242-260, 285-289).

3. Why might loss of function mutations emerge repeatedly in culture?

Authors note: We clarify in the Discussion that it is only for a few particular genes that loss-of-function mutants appear to be selected in culture, implying that it is only those few corresponding proteins for which absence leads to a higher mutation rate (297-312). We do not yet know the molecular or cellular mechanisms explaining each of these.

Reviewer #3 (Remarks to the Author):

This study analyzes clinical isolates from pediatric uncomplicated malaria infections in Ghana in continuous culture, showing that multiplication rates are associated with the initial parasite levels in patients. The authors study the genomic diversity within clinical isolates and present novel findings examining the role of genomic diversity and parasite growth in culture. Multiplication rates increased over time in culture, but were lower than rates of long-term culture-adapted strains, as has been previously shown. Interestingly, within-isolate genomic diversity decreased over time. Novel mutations did not appear to affect multiplication rates. At the first two time points, isolates with a single dominant genome sequence appeared to have a higher multiplication rate than isolates with multiple genome sequences.

This analysis provides insight into the role of genomics in parasite multiplication rates and suggests the need for further exploration of determinants of multiplication rates. It represents a novel investigation into a fundamental characteristic of malaria parasites. The reasoning is straightforward, and the experiments are presented in a clear, well-written fashion. The statistical analyses appear valid, and the approach reproducible.

MAJOR COMMENTS:

The use of the word “virulent” causes some confusion. Here, “virulent” appears to be used to indicate high parasitemia levels in patients. This is confusing, given that all isolates are from children with uncomplicated malaria. No isolates from severe malaria infections are used, which is a more typical context of mention of a “virulent parasite”. I suggest omitting the word “virulent” from the title.

Authors note: We now appreciate the terminology could be potentially confusing, as also pointed out by Reviewer 1. We have clarified this, and removed the term ‘virulent’ from the title as suggested (lines 1-2).

The Discussion mentions epigenetic regulation of transcriptional variation as a promising area of further investigation. Discussion of potential transcriptional analyses that could provide insight into multiplication rate differences would be helpful here as well. Are such analyses of RNA from these isolates underway?

Authors note: We have performed RNAseq analyses of the schizont-stage transcriptomes of a subset of these isolates (Tarr et al. BMC Genomics 2018), noting the importance of analysing

multiple independent experimental replicates of each isolate to obtain sufficient precision for comparisons. These showed some genes to differ in transcript levels between the cultured clinical isolates and long-term lab adapted isolates, and we have added this in the Discussion, although systematic analyses of transcription at all phases of the multiplication cycle have not been performed (lines 252-260). It is a good question that we hope will receive careful attention, as selective approaches may be needed given the large numbers of timepoints and experimental replicates that would ideally be performed for each isolate.

It would be beneficial to see a graph of multiplication rates for isolates with single genomes versus multiple genomes at each time point, ideally with the ability to see how rates for a given isolate tracks across time. The finding that genome diversity is not associated with multiplication rate at the third time point is interesting, and this graph would help to elucidate these associations over time.

Authors note: We appreciate this suggestion, as the data relating to this was only shown in separate supplementary tables. We have added a supplementary figure as suggested, which includes two different plots that show alternative ways of classifying the individual isolates (Supplementary Figure S2).

MINOR COMMENT:

Fig 3: A point of confusion: the text states that isolate 280 has a nonsense mutation at codon position 1422, whereas the figure states L285X.

Authors note: We are grateful for this error in the text being pointed out. The information in the figure is correct, and we have now corrected the text (line 167).

New Supplementary Figures and Table:

Supplementary Fig. S2. Comparison of multiplication rates in *P. falciparum* clinical isolates containing single genome sequences ($F_{WS} > 0.95$) or mixed genome sequences ($F_{WS} < 0.95$) at each of the timepoints of testing in culture. Each of the panels shows the isolate status in different ways, and numbers of isolates with multiplication rate assay and sequence data at each timepoint varies (all data on individual isolates are given in Table 1 and Supplementary Table 2). **A.** Each point represents an individual isolate, with 'days in culture' referring to the multiplication rate assay data only, so that black points represent isolates that had single genome sequences at *all* timepoints examined, white points represent those that had mixed genome sequences at *any* timepoint. **B.** Each point represents an individual isolate, shading of points representing whether they had single genome sequences (black) or mixed genome sequences (white) at each individual timepoint separately.

Supplementary Fig. S3. Numbers of merozoites per mature schizont counted in each of nine cultured clinical isolates and a test for correlation with multiplication rates at the final assayed timepoint. Counts of merozoites per mature schizont (with chemical blocking of egress using E64) were performed after different lengths of time in culture (closer to the middle or final assayed timepoints than to the first timepoint). **A.** Distributions of numbers of merozoites in individual schizonts. For each isolate, merozoites were counted in 100 schizonts (matured with chemical blocking of egress) as pooled counts from two different occasions (50 schizonts counted in each preparation). Moderate variation was seen, ranging from a mean of 16.5 for isolate 273 to 24.2 for isolate 293. **B.** Correlation between mean numbers of merozoites per schizont and multiplication rate assayed after day 153 of culture (Spearman's rho =0.46, P =0.20).

Supplementary Table S3. Gametocyte counts as a proportion of all parasite stages in the cultured *P. falciparum* clinical isolates

Patient Isolate	% Gametocyte counts in maintenance culture		
	Day 25	Day 77	Day 153
271	5 (60)	4 (332)	2 (125)
272	6 (52)	1 (409)	0 (234)
273	7 (110)	1 (276)	1 (220)
274	5 (100)	3 (289)	0 (397)
275	5 (76)	2 (311)	1 (128)
276	1 (86)	3 (392)	0 (318)
277	1 (82)	1 (254)	
278		1 (336)	6 (157)
279	7 (61)	2 (280)	
280	3 (89)	2 (439)	2 (202)
281		2 (311)	
282	10 (42)	2 (292)	0 (245)
283	0 (87)	3 (279)	0 (285)
284	9 (85)	4 (112)	2 (130)
285	7 (60)	3 (102)	1 (284)
286	2 (139)	2 (312)	1 (193)
287		2 (124)	0 (108)
288		0 (200)	0 (338)
289	1 (89)	1 (146)	1 (360)
290			2 (301)
291	10 (69)	0 (324)	1 (446)
292	2 (116)	0 (296)	1 (180)
293	9 (89)	2 (283)	0 (389)
294	5 (22)	3 (265)	0 (205)

Percentages of gametocytes are presented from parasite stage-differential counts made on the day and up to 7 days before or after when parasites were taken for each of the three timepoint assays. Numbers of parasites counted are shown in brackets. Blank cells correspond to points at which stage-differential counts were not performed on a given isolate.

REVIEWERS' COMMENTS:

Reviewer #1 (Remarks to the Author):
Please see attached PDF

Reviewer #2 (Remarks to the Author):

The authors have adequately addressed all of my previous concerns. This manuscript will make an important contribution to our understanding of the virulence phenotype in malaria parasites.

Reviewer #3 (Remarks to the Author):

I appreciate the careful consideration and response to reviewer queries.

The addition of Supplementary Figure S2 provides insight in terms of comparisons between single and mixed genome sequences at each time point. One additional visualization aspect that would be helpful would be a spaghetti plot to track how multiplication rates of individual isolates track across the three time points. This would require grouping all three time points for single genome isolates together and doing likewise for mixed genome sequences. It would then be possible to visualize consistencies in patterns across individual isolates over time for single versus mixed genome isolates.

Responses to Reviewers

We appreciate these highly informed and constructive reviews of the manuscript. We have made changes to revise accordingly, as noted and explained in bold for clarity below the individual comments.

In addition to the editing of text correspondingly, we have added two new Supplementary Figures, one new Supplementary Table (shown after the responses below), and four new references. Following editorial instructions, we specify the line numbers where changes and additions have been made. Note - the line numbers refer to those in the 'track changes' marked up version of the manuscript, sent as a separate pdf supplementary file to facilitate visible review – therefore the line numbers given do not correspond to those in the 'clean' revised manuscript when 'track changes' are not in view.

Reviewer #1 (Remarks to the Author):

The presented manuscript is well written and in line with previously reported data by the same group. The investigators propose that parasite multiplication rate is a determinant of Plasmodium falciparum virulence, intrinsically related to parasitemia levels of the patient at the time of parasite collection. To test this hypothesis the authors examined the multiplication rates in vitro of 24 isolates from Ghana over a period of ~5 months. The 3D7 clone, previously characterized by Murray et al, 2017, was used as control with a multiplication rate of ~8 fold/48 hours. Results suggest in vitro parasitemia levels associate with levels measured in vivo when each isolate was obtained. It also shows that isolate multiplication rates increase over time, a contrasting finding from the 2017 report by the same group. Finally, the work would benefit of further phenotypic characterization, including variation in merozoite numbers/schizont, erythrocyte invasion rate, knobs formation, and perhaps cell rosetting. Such information would provide further insights regarding the biological mechanisms of parasite growth and virulence proposed here.

Questions and suggestions:

Please define virulence and intensity of infection. This will help to give the readers a better picture of this biological aspect of malaria and frame the work presented.

Authors note: We agree with this, and have clarified that we are referring to intensity of infection and not any other potential virulence phenotype (lines 30, 41-42, 229, 324, referring to the 'Track changes' visible manuscript shown as a separate pdf as a supplementary file). We realise that by having placing a lot of the actual details of our measurements in the Supplementary Information we had accidentally sacrificed some clarity regarding the frame of what was done. While much needs to remain in Supplementary Information due to limited space, we now see that the details in the first table are really more suited as a main Table in the paper (we initially prepared the table for this purpose but at the point of original submission we opted for a more highly succinct and streamlined presentation), so we have reinstated this which should be more useful to readers (lines 639-647). We have altered the title to remove the word 'virulence' (lines 1-2), also in response to the suggestion of Reviewer 3.

Thank you for the answer, the changes improved the text and clarified the message.

Can other measurements such as number of merozoites/schizonts, erythrocyte invasion rate, knobs formation, and/or rosetting be obtained? Any of these further characterizations and comparisons between the 1st and last time points can provide new insights to the question of multiplication and virulence. It would be interesting to show if there is an association between multiplication rate and

number of merozoites/schizont or with capacity to invade erythrocytes. Knobs formation can easily be measured by gelatin flotation and it is known to reduce over time in culture.

Authors note: We agree that, given these findings, there are many components of parasite phenotypic variation at a cellular level that it would be interesting and relevant to study, as potentially involved in affecting the rate of multiplication of recently isolated parasites from clinical infections. We have extended the Discussion (lines 242-260) to include suggestions of the reviewer, and hope that many investigators will contribute to this important field, which is beyond the scope of any one research group. We have generated data on numbers of merozoites/schizont in counts on fully mature schizonts blocked from egress, and now include these in the Results (lines 182-194) and Methods (lines 422-434), with an additional supplementary figure (Supplementary Figure S3). We have also included counts of gametocytes (lines 196-209), with an additional supplementary table (Supplementary Table S3). The investigation and data inclusion in SupFigS3 is much appreciated.

Although, statistically, results from the 9 samples do not significantly explain most of the multiplication rate variation in 24 isolates, the trend suggests a positive correlation between number of merozoites and growth rate, which would make sense under the *assumption* that all merozoites have similar invasion capabilities. Is reference 7 appropriate when citing the neurosyphilis treatments from 1920-1950s?

Authors note: We are grateful for this query – the reference was accidentally misplaced, and has been moved to the following sentence in the Introduction (line 62).

Happy to help!

The higher multiplication rates of isolates at the end of the experiment suggest the selection of a subpopulation of parasites better equipped to grow under these conditions. If this is the case, please explain the different findings by Murray et al 2017: "Four of the clinical isolates cultured for longer periods were assayed at three different time points after culture initiation (up to 76 or 100 days for each isolate). These showed no significant changes in multiplication rate over time in culture."

Authors note: We have now added reference to this in the Discussion (lines 290-295). The study by Murray et al. 2017 was not well powered to see changes over time, as only four isolates were cultured for periods beyond the first month, whereas the current study of 24 isolates cultured over 5 months had sufficient data to be able to statistically detect changes.

Thanks for including this in the text, it helps as such questions may be anticipated by those in field.

From results: "Twenty-four new clinical isolates were tested in exponential multiplication rate assays after different lengths of time of continuous laboratory culture, when parasites could be effectively tested in triplicate with erythrocytes from three different donors."

Please clarify and provide detailed information of previous time of continuous culture for each isolate. How does time to become "testable" relate to multiplication rate and is there information on the genotype of the sample directly obtained from the patient?

Authors note: We have now made this clearer (lines 100-103, 357, 361-362). All isolates were cultured in parallel at the same time, using exactly the same conditions. No isolates were pre-cultured before thawing in the laboratory on day 0, as the cryopreserved blood samples were directly from patients. As we intentionally did not investigate the phenotypes until day 25 (by which time all isolates were growing in the same donor cells which avoided possible confounding variation that could have occurred if measurements were done earlier due to heterogeneous patient erythrocytes) we did not systematically analyse parasite sequences before day 25, although the average within-isolate diversity was similar to what we had previously seen when sequencing samples directly from patients (lines 265-269).

Line 364: "of" can be removed from: "Briefly, 12% NaCl (0.5 times the original volume) of was added dropwise..."

The inclusion of these details in the Methods are much appreciated.

The authors suggest that the variability in asexual multiplication rates observed here cannot be explained by the rates of switching from asexual replication to sexual differentiation and cite reference 26. Were gametocytes observed in the patient sample? Please note if gametocytes were

ever observed during the in vitro cultivation. Discuss how parasite multiplication rate, virulence, and transmission may be related.

Authors note: We appreciate the query, and have now presented data on gametocyte counts during the culture of each of the individual isolates, focusing on two-week windows of time centred on the timepoints at which the cultures were sampled for the exponential growth rate assays. We have summarised these results in the Results text (lines 196-209), and added a new supplementary table (Supplementary Table S3). Consistent with the previous interpretation, the proportions of gametocytes were low in all cultures, and could not explain the substantial variation in multiplication rates. Although the interpretation has not changed, we have referred to this in the Discussion (lines 237-240).

Beautiful. The data included clearly supports the authors suggestion and previous observations. Thank you for looking into it and adding the information.

Methods: was PCR for other human malaras performed? Could the presence of other malaras affect the initial adaptation?

Authors note: PCR was not performed for other human malaria parasite species, as these do not grow in continuous culture using under the methods here, so would not be present in culture at any of the times of assay. The other species are usually only present in a minority of infections in this area, at low levels in the blood. We now specify that microscopy identified one of the patient samples (number 290) to have had *P. malariae* in the blood along with *P. falciparum* at time of collection (lines 344-346), but no isolate had this or any other species identified alongside *P. falciparum* during culture.

Great. It makes sense, and the *P. malariae* type of information is exactly what I was wondering. Such studies can add to the limited information about malarial infections with multiple species. Thank you.

Trager W and Jensen JB on 1976, cited by the authors in the methods, does not describe how to thaw parasites cryopreserved with glycerolyte. Considering this work is highly dependent on the cultivation methods and the fact that currently most laboratories use modified conditions from the 1976 publication, I think it is important to describe a few further details: the size of cryopreserved samples, if the cryopreserved material came directly from patients or from short-term cultivations, steps to thaw, etc...

Authors note: We agree that it is appropriate to give these details, and have added these now to the Materials and Methods (lines 362-373).

The Methods section is much improved and can easily be followed step by step.

The group uses 3 different erythrocyte donors during the experiments, was haemoglobin genotyped? Is there information regarding alpha thalassemia prevalence among the donors? These can impact in vitro growth and therefore should be addressed.

Authors note: Theoretically if such erythrocyte variants were present in cultures they could influence growth, but all the erythrocyte donors for the assays were laboratory staff of LSHTM in London among which the general frequency of haemoglobin variants or thalassaemia is very low (lines 383-386), and all assays done in cells from triplicate donors included the long-term laboratory parasite clone 3D7 as a control, which consistently gave the multiplication rate of approximately 8-fold per 48 hours (lines 411-413).

Great, blood source details can be important and acknowledgment of such possible variations should be considered.

“The long-term laboratory adapted *P. falciparum* clone 3D7 was assayed in parallel as a control in all assays, consistently showing a multiplication rate of approximately 8.0 fold per 48 hours as described previously 19” Should this be expected? Can’t the multiplication rate increase over time

for clones as well?

Authors note: We see a stable multiplication rate of clone 3D7 over time as described in this manuscript and also in the earlier work of Murray et al. 2017. We cannot exclude the possibility that some long-term clones could vary in multiplication rates, but we have seen consistent rates for long-term laboratory clones in exponential growth assays (with clone Dd2 being approximately 10-fold per 48 hours, while HB3 and D10 are approximately 8-fold, as described by Murray et al. 2017).

Good. Thank you for carefully addressing all my questions. The added details and data enriched the report. I have no further suggestions.

Reviewer #2 (Remarks to the Author):

It is known that Plasmodium falciparum disease severity is correlated with the number of blood stage parasites. The authors explore the relationship between intrinsic multiplication rate variation of P. falciparum in laboratory cultures, and the original in vivo parasitemia and clinical presentation for patient samples from a highly endemic area in West Africa. Previous studies measuring the relationship between multiplication rates and patient disease severity have yielded conflicting and inconclusive results. The current study is carefully designed to avoid both host response and sample handling variables; and parasite isolates were allowed to grow in culture for a period of weeks during which time multiplication rates were periodically measured. Both technical and biological replicates are included, with quantitation over multiple cycles and assay replicates using unrelated donor erythrocytes. The methodology used in this study has been previously published by this group. The current study differs from the previous one in that they link in vitro replication rates back to parasite levels in patients at clinical presentation. The authors found that not only are intrinsic multiplication rates inherently stable in laboratory culture as they had shown previously, but that they were significantly and consistently positively correlated with blood parasitemia levels measured in vivo. This is an interesting observation that adds to our understanding of the parasites' contribution to the virulence phenotype. The carefully controlled experiments provide confidence that the observations are robust and likely broadly applicable to other isolates and to our understanding of virulence in P. falciparum. The statistical analyses are appropriate and the work is well described and reproducible.

1. Laboratory culture conditions are not nutrient-limited and thus one would not expect to see alterations in replication rates due to competition between co-cultured strains. However, the nutrient environment in vivo is likely to differ considerably from laboratory conditions. Would you expect multiplication rates to be altered in response to competing parasites in the host?

Authors note: Potential interactions between parasites within the host may be more complex than in the artificial culture system, either due to varying nutrient environment or other host-related variables, so we agree that there may be interactions in vivo that could lead to modifications of multiplication rates and have added to the Discussion to note this (lines 262-265).

2. What explanation can be given for mean multiplication rates increases over time for in vitro parasite cultures? Adaptation to culture conditions? What are the implications for interpreting biological significance of multiplication rates in long term cultures? It seems as though a very narrow window of time exists within which parasite phenotypes can be assessed in culture.

Authors note: These are good questions which may be commented on while they remain open to some extent, so we have added to the Discussion accordingly (lines 242-260, 285-289).

3. Why might loss of function mutations emerge repeatedly in culture?

Authors note: We clarify in the Discussion that it is only for a few particular genes that loss-of-function mutants appear to be selected in culture, implying that it is only those few corresponding proteins for which absence leads to a higher mutation rate (297-312). We do not yet know the molecular or cellular mechanisms explaining each of these.

Reviewer #3 (Remarks to the Author):

This study analyzes clinical isolates from pediatric uncomplicated malaria infections in Ghana in continuous culture, showing that multiplication rates are associated with the initial parasite levels in patients. The authors study the genomic diversity within clinical isolates and present novel findings examining the role of genomic diversity and parasite growth in culture. Multiplication rates increased over time in culture, but were lower than rates of long-term culture-adapted strains, as has been previously shown. Interestingly, within-isolate genomic diversity decreased over time. Novel mutations did not appear to affect multiplication rates. At the first two time points, isolates with a single dominant genome sequence appeared to have a higher multiplication rate than isolates with multiple genome sequences.

This analysis provides insight into the role of genomics in parasite multiplication rates and suggests the need for further exploration of determinants of multiplication rates. It represents a novel investigation into a fundamental characteristic of malaria parasites. The reasoning is straightforward, and the experiments are presented in a clear, well-written fashion. The statistical analyses appear valid, and the approach reproducible.

MAJOR COMMENTS:

The use of the word “virulent” causes some confusion. Here, “virulent” appears to be used to indicate high parasitemia levels in patients. This is confusing, given that all isolates are from children with uncomplicated malaria. No isolates from severe malaria infections are used, which is a more typical context of mention of a “virulent parasite”. I suggest omitting the word “virulent” from the title.

Authors note: We now appreciate the terminology could be potentially confusing, as also pointed out by Reviewer 1. We have clarified this, and removed the term ‘virulent’ from the title as suggested (lines 1-2).

The Discussion mentions epigenetic regulation of transcriptional variation as a promising area of further investigation. Discussion of potential transcriptional analyses that could provide insight into multiplication rate differences would be helpful here as well. Are such analyses of RNA from these isolates underway?

Authors note: We have performed RNAseq analyses of the schizont-stage transcriptomes of a subset of these isolates (Tarr et al. BMC Genomics 2018), noting the importance of analysing

multiple independent experimental replicates of each isolate to obtain sufficient precision for comparisons. These showed some genes to differ in transcript levels between the cultured clinical isolates and long-term lab adapted isolates, and we have added this in the Discussion, although systematic analyses of transcription at all phases of the multiplication cycle have not been performed (lines 252-260). It is a good question that we hope will receive careful attention, as selective approaches may be needed given the large numbers of timepoints and experimental replicates that would ideally be performed for each isolate.

It would be beneficial to see a graph of multiplication rates for isolates with single genomes versus multiple genomes at each time point, ideally with the ability to see how rates for a given isolate tracks across time. The finding that genome diversity is not associated with multiplication rate at the third time point is interesting, and this graph would help to elucidate these associations over time.

Authors note: We appreciate this suggestion, as the data relating to this was only shown in separate supplementary tables. We have added a supplementary figure as suggested, which includes two different plots that show alternative ways of classifying the individual isolates (Supplementary Figure S2).

MINOR COMMENT:

Fig 3: A point of confusion: the text states that isolate 280 has a nonsense mutation at codon position 1422, whereas the figure states L285X.

Authors note: We are grateful for this error in the text being pointed out. The information in the figure is correct, and we have now corrected the text (line 167).

New Supplementary Figures and Table:

Supplementary Fig. S2. Comparison of multiplication rates in *P. falciparum* clinical isolates containing single genome sequences ($F_{WS} > 0.95$) or mixed genome sequences ($F_{WS} < 0.95$) at each of the timepoints of testing in culture. Each of the panels shows the isolate status in different ways, and numbers of isolates with multiplication rate assay and sequence data at each timepoint varies (all data on individual isolates are given in Table 1 and Supplementary Table 2). **A.** Each point represents an individual isolate, with 'days in culture' referring to the multiplication rate assay data only, so that black points represent isolates that had single genome sequences at *all* timepoints examined, white points represent those that had mixed genome sequences at *any* timepoint. **B.** Each point represents an individual isolate, shading of points representing whether they had single genome sequences (black) or mixed genome sequences (white) at each individual timepoint separately.

Supplementary Fig. S3. Numbers of merozoites per mature schizont counted in each of nine cultured clinical isolates and a test for correlation with multiplication rates at the final assayed timepoint. Counts of merozoites per mature schizont (with chemical blocking of egress using E64) were performed after different lengths of time in culture (closer to the middle or final assayed timepoints than to the first timepoint). **A.** Distributions of numbers of merozoites in individual schizonts. For each isolate, merozoites were counted in 100 schizonts (matured with chemical blocking of egress) as pooled counts from two different occasions (50 schizonts counted in each preparation). Moderate variation was seen, ranging from a mean of 16.5 for isolate 273 to 24.2 for isolate 293. **B.** Correlation between mean numbers of merozoites per schizont and multiplication rate assayed after day 153 of culture (Spearman's rho =0.46, P =0.20).

Supplementary Table S3. Gametocyte counts as a proportion of all parasite stages in the cultured *P. falciparum* clinical isolates

Patient Isolate	% Gametocyte counts in maintenance culture		
	Day 25	Day 77	Day 153
271	5 (60)	4 (332)	2 (125)
272	6 (52)	1 (409)	0 (234)
273	7 (110)	1 (276)	1 (220)
274	5 (100)	3 (289)	0 (397)
275	5 (76)	2 (311)	1 (128)
276	1 (86)	3 (392)	0 (318)
277	1 (82)	1 (254)	
278		1 (336)	6 (157)
279	7 (61)	2 (280)	
280	3 (89)	2 (439)	2 (202)
281		2 (311)	
282	10 (42)	2 (292)	0 (245)
283	0 (87)	3 (279)	0 (285)
284	9 (85)	4 (112)	2 (130)
285	7 (60)	3 (102)	1 (284)
286	2 (139)	2 (312)	1 (193)
287		2 (124)	0 (108)
288		0 (200)	0 (338)
289	1 (89)	1 (146)	1 (360)
290			2 (301)
291	10 (69)	0 (324)	1 (446)
292	2 (116)	0 (296)	1 (180)
293	9 (89)	2 (283)	0 (389)
294	5 (22)	3 (265)	0 (205)

Percentages of gametocytes are presented from parasite stage-differential counts made on the day and up to 7 days before or after when parasites were taken for each of the three timepoint assays. Numbers of parasites counted are shown in brackets. Blank cells correspond to points at which stage-differential counts were not performed on a given isolate.